# Computational Language Acquisition with Theory of Mind

**Andy Liu**
Harvey Mudd College
Claremont, CA, USA
{ajliu}@g.hmc.edu

**Hao Zhu, Emmy Liu, Yonatan Bisk, Graham Neubig**
Language Technologies Institute
Carnegie Mellon University
Pittsburgh, PA, USA
{zhuhao, mengyan3, ybisk, gneubig}@cs.}cmu.edu

## Abstract

Unlike current state-of-the-art language models, young children actively acquire language through interactions with their surrounding environment and caretakers. One mechanism that has been argued to be critical to language learning is the ability to infer the mental states of other agents in social environments, coined Theory of Mind (ToM) by Premack & Woodruff (1978). Drawing inspiration from the modern operationalized versions of ToM implemented in Rabinowitz et al. (2018) and Zhu et al. (2021), we build language-learning agents equipped with ToM, and measure its effects on the learning process. We model ToM by giving the speaker agent an internal listener model that is trained alongside the speaker and used to rerank potential utterances. We experiment with varying task difficulty, hypothesizing that models will acquire more complex language to adapt to stronger environmental pressures. We find that training speakers with a highly weighted ToM listener component leads to performance gains in our image referential game setting. We also find some evidence that increasing task difficulty in the training process results in more fluent and precise utterances in evaluation. This suggests the potential utility of further incorporating ToM, as well as other insights from child language acquisition, into computational models of language acquisition[1].

## 1 Introduction

Human languages are fundamentally shaped by social-communicative goals in the grounded world. Modern theories from developmental psychology often attribute humans' unique ability to quickly acquire and adapt language to their ability to ascribe mental states to other agents (Tomasello, 2005), an ability also known as Theory of Mind (ToM).

Some previous studies have attempted to perform computational modeling of ToM. For instance, ToM-like mechanisms have been demonstrated to allow models to better predict the behavior of a future agent (Rabinowitz et al., 2018), model agents' beliefs in a negotiation (Cao et al., 2018) or a cooperative game (Bard et al., 2020), or choose good utterances based on the listener's linguistic abilities (Zhu et al., 2021). However, the effects of ToM have not yet been studied in the higher-level context of computational language acquisition.

In this paper, we study how an *internal ToM mechanism* and *external environmental pressure* contribute to language learning. We use an image referential game setting consisting of a series of training episodes between a speaker, which represents a language learner (Zhu et al., 2022), and a listener, which represents a fluent teacher. When presented with a set of images, one of which is the target referent, the speaker must learn to generate an English utterance that the listener can use to select the target. The speaker is rewarded for generating utterances that are used to correctly guess the target image. Additionally, the speaker may be given feedback depending on the confidence the listener has in the selection. This setting provides an attractive test-bed for testing the effects of various reward signals or model designs on the speaker's learned language; previous studies of pragmatics in language acquisition, such as Andreas & Klein (2016), have used similar settings.

---

[1]Code and data can be found at `https://github.com/neulab/ToM-Language-Acquisition`.

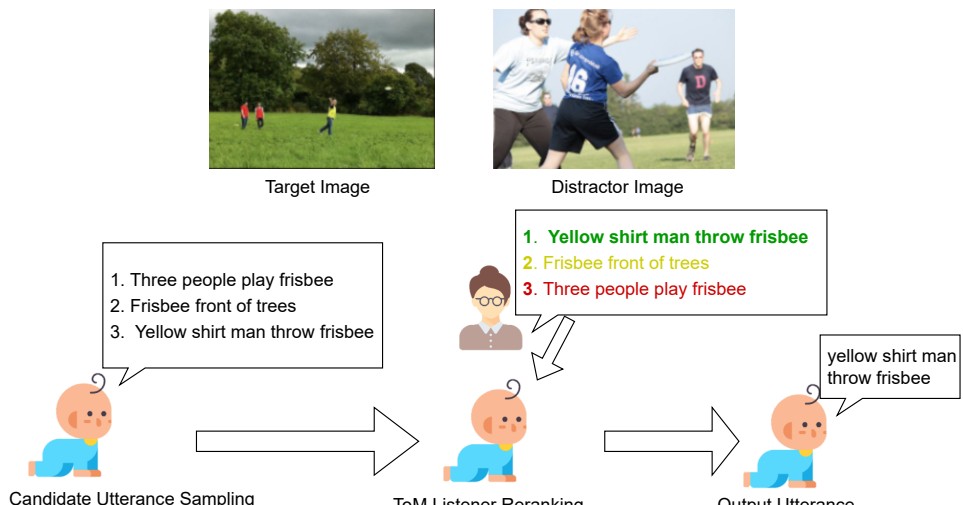

Figure 1: An example of how ToM is used by our speaker models in our implementation. The speaker first generates candidate utterances before reranking them according to the probability given to each target image, utterance pair by the internal ToM listener. It then selects either the highest-scoring or a random utterance from the candidate pool to output.

Within this setting, we seek to better understand how models with and without ToM adapt to their environments. We focus on two specific research questions in this area:

RQ1. How does the inclusion of ToM in language acquisition speaker models affect their performance and learned language? (*internal ToM mechanism*)

RQ2. How do our models adapt to more difficult referential game environments during the language acquisition process? (*external environmental pressure*)

We study the impact of ToM (RQ1) by modeling an internal listener module within our speakers that aims to predict which utterances are most likely to result in the desired listener behavior. By incorporating the probabilities given to the target image by the ToM listener into the utterance reranking process, as shown in Fig. 1, we select for more pragmatic utterances sampled from the speaker's distribution. To study the impact of environmental pressure (RQ2) on our speakers, we create referential games with different difficulties by sampling distractors from different distributions. These distributions are based on the similarity between images calculated by various image representation models, including CLIP (Radford et al., 2021), RoBERTa (Liu et al., 2020), and TF-IDF variants.

In experiments, we find that (RQ1) speaker models including ToM components generally outperform those that do not in terms of fluency and final accuracy. We also find that (RQ2) training with more visually and semantically similar distractor referents causes the speaker model to develop *longer, more fluent, and more precise* utterances to distinguish between potential referents, although these do not always translate to gains in referential game performance. These results suggest contributions to language acquisition from both ToM and environmental pressures in this setting. We still find significant gaps between the language that our speaker model acquires and human captions. Additionally, we restrict both the vocabulary and the maximum length of our speakers' utterances. These both suggest that there is still room for improvement in this class of models. However, we hope our results still hint at both better training methods for pragmatic language models and a deeper computational understanding of human language acquisition.

## 2 IMAGE REFERENTIAL GAME ENVIRONMENT

Following Lazaridou et al. (2016); Lowe et al. (2019); Zhu et al. (2022), we consider image referential games with real world images, where a speaker and a listener collaborate on identifying a target image $x \in C$ among distractors randomly sampled from a set of candidates $C$. The identity of the target is known only to the speaker. The speaker generates an English-language utterance $u$

that describes the target image, which is then passed to the listener. The listener will either select an image $\hat{x}$, in the case where they understand the utterance with a high enough probability or refuse to act, in the case where they do not. The listener can also determine whether or not to provide feedback (linguistic input in the form of the ground-truth caption of the target image) to the speaker.

Speaker training is motivated by a reward function that seeks to model the communicative goal of directing the listener to the correct referent. The speaker gets a positive reward of 1 if the listener chooses the target image and a penalty of $-1$ if the listener chooses the wrong target image. Additionally, a smaller penalty $w_{\texttt{noop}}$ is applied to the speaker if the listener chooses to not choose an image; this is done to penalize the speaker for generating unclear utterances.

## 2.1 Agent Formulation

We follow the speaker and listener formulations defined in Zhu et al. (2022). The speaker is a model $f : I \rightarrow \Sigma^*$ that generates utterances (sequences of tokens from its vocabulary) that correspond to a provided image. We define $\Sigma^*$ as the speaker vocabulary and $I$ as the set of candidate images.

The listener is defined on $\Sigma^*$ and $I^N$, which is a set of $N$ candidate images sampled from $I$. Given an utterance $\in \Sigma^*$ and an observation space $\in I^N$, the listener will either return the index of its predicted image or $\texttt{noop}$, if it cannot predict an image. It may also return the ground-truth caption from its vocabulary. We define it with the model $g : \Sigma^* \times I^N \rightarrow ([0, 1, \ldots N - 1] \cup \texttt{noop}) \times \Sigma^*$.

## 2.2 Speaker Design

The speaker that we use is a captioning model composed of a pretrained ResNet (He et al., 2016) model and an LSTM-based utterance generation model. After generating an embedding vector of the target image using the ResNet model, the speaker autoregressively generates an utterance $u = \{u_i\}_{i=1}^{M}$ using the LSTM network:

$$
\begin{aligned}
&P(u_i \mid u_1, u_2, \ldots, u_{i-1}, x) \\
&\propto \exp(w_{u_i}^T \texttt{LSTM}(w_{u_1}, w_{u_2}, \ldots, w_{u_{i-1}}, h_0 = \texttt{ResNet}(x))),
\end{aligned}
\tag{1}
$$

where $w_{u_i} \in \mathbb{R}^{d_w}$ is the word embedding of $u_i$. The speaker generates a sequence of tokens to either the maximum length, which is set to 20 in our experiments, or an end-of-sequence token. In our experiments, we use a vocabulary size of 200 to limit the size of the action space and to more realistically mimic the vocabulary of a language learner.

## 2.3 Listener Design

Given a set of candidate images $I_j$ ($j \in 1, 2 \ldots n + 1$), where $n$ is the number of distractors, and an utterance $u$, the listener computes embeddings for each image, $L(I_j)$, for the utterance, $L(u)$. It then takes the dot product of each $L(I_j)$ with $L(u)$, and the softmax of the dot products to compute the probability $P(I_j|u) \propto \exp(L(I_j)L(u))$ of the utterance referring to image $I_j$ for each image.

The listener also has a rule-based component to control when to give linguistic input in the form of the ground-truth caption $C(x)$. We introduce two thresholds, $\theta_1$ and $\theta_2$. After the listener computes the most likely target image, $t = \arg\max P(I_j|u)$, and the probability given to this choice, $P_{\max} = \max P(I_j|u)$, it returns its choice $\hat{x}$ and input $f_{\texttt{listener}}$ according to the following logic:

$$
(\hat{x}, f_{\texttt{listener}}) = \begin{cases} (\texttt{noop}, 0) & P_{\max} < \theta_1 \\ (t, C(x)) & \theta_1 < P_{\max} < \theta_2 \\ (t, 0) & P_{\max} > \theta_2 \end{cases}
\tag{2}
$$

This control strategy mimics a caregiver who only gives feedback when they understand what the language learner is referring to, but wants to direct it to generate higher-quality utterances. When the listener's confidence is very low, it will neither select a target image nor provide linguistic input, as the speaker utterance is too low-quality. Meanwhile, when the listener's confidence is very high, it will stop giving linguistic input, as further improvement is deemed unnecessary.

We use an LSTM-based model to learn embeddings of speaker utterances combined with a ResNet model to derive image features. Because we want our listener to at least model a competent user of the language that the speaker is trying to acquire, some pretraining is required before initializing a listener in a new environment. The controller values $\theta_1, \theta_2$ and the ResNet parameters are set after preliminary experimentation on the dataset. The listener's language network parameters are then trained with mini-batch stochastic gradient descent to optimize the network values

$$\theta_{\text{listener}} = \arg\max_{\theta} \mathbb{E}_{C \sim \mathcal{U}(I)^N} \frac{1}{N} \sum_{i=1}^{N} \log P_{\text{listener}}(i \mid U_i^*, C; \theta) \tag{3}$$

## 3 SPEAKER TRAINING PROCESS

### 3.1 COMMUNICATIVE GOALS

We model communicative goals and learning from linguistic input as two separate learning objectives for our speaker network, which we combine with an adjustable coefficient $\lambda$. The communicative goal reward $\mathcal{O}_{CG}$ is simply the average reward (defined in §2) over each game in the training episode. The space of possible output utterances is discrete and non-differentiable, so our speaker learns its policy $\pi$ via the reinforcement learning method PPO (Schulman et al., 2017).

### 3.2 LEARNING FROM LINGUISTIC INPUT

We model the learning from linguistic input objective as the maximum likelihood of the listener input in the speaker models and optimize this continuous function with stochastic gradient descent.

$$\mathcal{O}_{LI} = \mathbb{E}_{x,C,u \sim \pi(u|x)} \log \pi(U_{f_{\text{listener}}(u,C)}^* \mid x, C) \tag{4}$$

We then combine the separate learning objectives for each task using a coefficient $\lambda \in [0,1]$:

$$\mathcal{O}_{\text{joint}} = \lambda \mathcal{O}_{CG} + (1 - \lambda) \mathcal{O}_{LI} \tag{5}$$

Next, we introduce ToM Modeling, and an additional task objective, $\mathcal{O}_{ToM}$. This is a cross-entropy objective that represents how accurate the speaker's internal listener model is. Similar to the objective above, we combine these into a single learning objective for the entire speaker network.

## 4 INTRODUCING ToM MODELING

We incorporate into the speaker architecture an LSTM-based ToM listener. This listener model is trained alongside of the rest of the speaker network to learn the "mental state" of the actual listener – the neural network parameters that allow it to best replicate the probabilities that the listener would assign to image-utterance pairs. It uses a similar architecture to the actual listener, combining its own learned sentence embeddings from an LSTM network with pretrained image embeddings using ResNet. In other words, our ToM listener seeks to learn the sentence embedding model $\theta_{ToM}$ that will maximize the probabilities assigned to the listener's choice given an utterance.

$$\theta_{ToM} = \arg\max_{\theta} P_{ToM}(u_i|\hat{x}) \propto \exp(\theta_{ToM}^T(u_i) \cdot \texttt{ResNet}(\hat{x})) \tag{6}$$

We introduce ToM into the speaker model by having our speaker sample candidate utterances and rerank them with the help of the ToM listener. Our ToM speaker first samples $N$ candidate utterances $U = \{u^{(i)}\}_{i=1}^N$ from its distribution. Next, we generate a speaker and a listener score for each:

$$P_{\texttt{speaker}}(u) \propto \prod_i P(u_i|u_1, u_2, \ldots u_{i-1}, x) \tag{7}$$

$$P_{\texttt{ToM}}(x|u) \propto \exp(\texttt{LSTM}^T(u)\texttt{Resnet}(x)) \tag{8}$$

Here, the probability $P(u_i|u_1, u_2, \ldots u_{i-1}, x)$ is computed according to 1. We then combine these scores using a listener weight hyperparameter $w_l$ to get the overall score assigned to each utterance. Finally, we select the best utterance, $u^b$, as the argmax of this score:

$$u^b = \arg\max_U (P_{\texttt{ToM}}(x|u^j)^{w_l} \cdot p_{\texttt{speaker}}(u^j)) \tag{9}$$

In our experiments, we train models with three different settings of $w_l$. We train models with $w_l = 0$ to isolate the effects of the ToM listener from the effects of the new model architecture. We train models with $w_l = 1$ which weigh the speaker and ToM listener input equally. Finally, we train models where $w_l$ is the arbitrarily high constant 1000. In this case, the listener score dominates, and our speaker essentially seeks to maximize $P_{\texttt{listener}}(x|u^i)$, which it approximates with the ToM listener score. If we replace the learned ToM listener with a copy of the external listener, this utterance sampling process is close to that of the learned Rational Speech Act (RSA) model in Andreas & Klein (2016), which also trains pragmatic speakers on referential game environments. We compare our ToM speakers to these "RSA" speakers to evaluate the impact of using our learned approximation rather than the actual listener.

Finally, we introduce a hyperparameter $\sigma$: our speaker outputs a random utterance with probability $\sigma$ and $u^b$ with probability $1 - \sigma$. $\sigma$ is set to decay linearly over time; this randomization is done to promote early exploration.

By default, we begin with an untrained listener that will be trained to emulate the actual listener's outputs over time. To train the ToM listener, we introduce a third training objective in addition to $\mathcal{O}_{LI}$ and $\mathcal{O}_{CG}$, the ToM objective $\mathcal{O}_{ToM}$, defined as the cross-entropy loss between the distribution of the ToM listener and that of the true listener. Thus, we are training it to give higher probabilities to the listener's choice $\hat{x}$ based on the speaker's utterances. Formally,

$$\mathcal{O}_{ToM} = \begin{cases} -\log P_{ToM}(\hat{x}|u) & P_{\max} > \theta_1 \\ 0 & P_{\max} < \theta_1 \end{cases} \tag{10}$$

Note that this requires a choice to be made by the listener, so we mask out the case where the actual listener does not choose an image by setting it to zero. This is done using the same parameter $\theta_1$ and listener confidence $P_{\max}$ that were used to control linguistic input in 2. We add this to the joint learning objective previously defined in 5 to compute our combined objective for the ToM speaker system as a whole:

$$\mathcal{O}_{\text{joint}} = \lambda\mathcal{O}_{CG} + (1 - \lambda)\mathcal{O}_{LI} + \mathcal{O}_{ToM} \tag{11}$$

However, an untrained listener can introduce significant noise to the utterance generation process. To counteract this, we anneal the influence of the listener, by linearly increasing $w_l$ from 0 to the final $w_l$ value over a fixed number of steps. This allows our speaker to only begin using its ToM listener when the listener is at least somewhat capable.

## 5 INCREASING DISTRACTOR DIFFICULTY

Previous studies on language acquisition have found that infants initially look at familiar objects when hearing semantically related labels for different objects, but adapt to more difficult tasks over time by learning more complex semantic representations of objects (Bergelson & Aslin, 2017a). Additionally, Yuksekgonul et al. (2022) showed that contrastive training on harder distractors improved visiolinguistic models' performance on tasks involving compositionality. We hypothesize that the usage of more similar distractor images might similarly force our speaker to generate more complex utterances due to the need to further distinguish between images. This motivated us to generate more similar distractor images in the training process in order to achieve such an effect. To do so, we computed a similarity rank between images based on visual and semantic similarity. Then, after selecting a "target" image in the training process, we sampled images with high similarities to the target image to use as distractors during training.

We experiment with three options for calculating image/caption similarity:

| Original Image | Random Distractor | Image Similarity | Caption Similarity | Hybrid Similarity |
|---|---|---|---|---|

*a little girl holding a kitten next to a blue fence.* / *two sheep standing next to each other in the snow.* / *a little dog sitting on a wooden bench.* / *a woman hugs a gray cat to her chest.* / *a kitten that is sitting down by a door.*

Figure 2: An example image-caption pair, and distractors ranked as similar by various metrics. Caption and Hybrid similarities are computed with TF-IDF - weighted RoBERTa embeddings.

- **Visual Cosine Similarity.** We compute the most visually similar images based on the cosine similarity of image embeddings from a pretrained ResNet model. For each image, we save a similarity ranking of the next 1000 most visually similar images, and then select distractors from this set during training whenever the original image is selected as a target.
- **Textual Cosine Similarity.** We calculate the textual similarity of captions by taking the cosine similarity of vector representations of the text. We experiment with both (1) dense vectors using embeddings calculated using either the pretrained RoBERTa or the pretrained CLIP mean pooled models from the sentence-transformers library (Reimers & Gurevych, 2019), and (2) sparse vectors using one-hot vectors of each word. We also experimented with using term frequency – inverse document frequency (TF-IDF) to weight the vector for each token in our captions when computing caption similarity as a way to upweight rarer content words.
- **Visual+Textual Similarity.** In cases where both image and caption similarity were used, we added a caption weight parameter $w_c \in [0, 1]$ and used it to compute a weighted average of image and caption similarity. We hoped that using hybrid methods ($w_c = 0.5$) would capture distractors with both visual and semantic similarities.

We also train models on randomly selected, or "easy", distractors to create a baseline to study the effects of distractor difficulty. Examples of a selection of the distractor settings are shown in Fig. 2.

# 6 EXPERIMENTAL SETUP AND RESULTS

## 6.1 EXPERIMENTAL SETUP

We focus on two main results: A speaker's ability to use language to correctly discriminate between the target and distractor images, as well as the quality of the language speakers use to do so.

To evaluate the former, we primarily consider average accuracy, or how often the listener selects the correct referent. We evaluate the quality of our speakers' learned languages using fluency, average length, F1 score for salient parts of speech, and caption quality scores. We use a fluency score that measures the grammatical quality of output utterances (Kann et al., 2018). This is defined as the average gain in log probability when moving from a unigram model to a pretrained language model.

$$\text{fluency} = \frac{1}{|u|}(\ln(p_M(u)) - \ln(p_U(u))) \tag{12}$$

We train a unigram model, $p_U$, and fine-tune GPT-2 large (Radford et al., 2019), $p_M$, on our training set.

In addition to fluency and accuracy, we consider two additional performance metrics. To better evaluate the overall caption quality of the system, we compute BLEU score (Papineni et al., 2002) using the implementation found in Bird et al. (2009). Additionally, to evaluate the accuracy of the learned listener model, we compute ToM Accuracy, defined as how often the candidate image that the ToM listener believes has the highest probability of being chosen is the actual listener's choice.

We also consider part-of-speech statistics to evaluate utterance quality. Parts of speech for words in speaker utterances and ground-truth captions are tagged with spaCy (Honnibal et al., 2020). We use

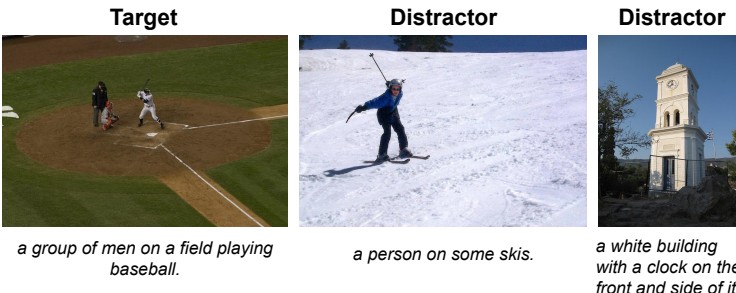

| **Target** | **Distractor** | **Distractor** |
|---|---|---|
| *a group of men on a field playing baseball.* | *a person on some skis.* | *a white building with a clock on the front and side of it.* |

**Base Speaker:** A baseball player holding a holding umbrella.
**Speaker Trained on Hard Distractors:** A baseball player holding a bat on a field.
**ToM Speaker Trained on Hard Distractors:** A group of men are playing baseball outside.

Figure 3: Examples of output utterances generated by various speaker models. Here, the "Hard Distractors" referenced are selected with hybrid metrics computed with CLIP embeddings. Both training on more difficult distractors and training with ToM leads to more accurate, fluent utterances.

Table 1: Performance and language features of various ToM speakers.

| Model ToM Weight | Distractors | Performance Acc | BLEU | Fluency | ToM | POS F1 ADJ | ADP | NOUN | VERB |
|---|---|---|---|---|---|---|---|---|---|
| Baseline (No ToM) | Easy | 0.81 | 0.20 | 1.50 | N/A | 0.16 | 0.52 | 0.41 | 0.38 |
| Baseline (No ToM) | Hard | 0.81 | 0.24 | 1.87 | N/A | 0.24 | 0.58 | 0.46 | 0.45 |
| Gold Standard | N/A | 0.92 | 1.00 | 2.52 | N/A | 1.00 | 1.00 | 1.00 | 1.00 |
| Zero | Hard | 0.83 | 0.26 | 1.99 | 0.81 | 0.22 | 0.64 | 0.49 | 0.47 |
| Normal | Hard | 0.85 | 0.26 | 2.25 | 0.88 | 0.22 | 0.65 | 0.52 | 0.49 |
| **High** | **Hard** | **0.88** | **0.27** | **2.23** | **0.89** | **0.22** | **0.66** | **0.52** | **0.50** |
| High RSA | Hard | 0.87 | 0.28 | 2.26 | 0.93 | 0.23 | 0.65 | 0.50 | 0.49 |
| Zero | Easy | 0.85 | 0.25 | 1.73 | 0.85 | 0.21 | 0.57 | 0.48 | 0.49 |
| Normal | Easy | 0.88 | 0.26 | 2.09 | 0.91 | 0.21 | 0.64 | 0.50 | 0.52 |
| **High** | **Easy** | **0.88** | **0.27** | **2.07** | **0.91** | **0.22** | **0.65** | **0.51** | **0.50** |
| High RSA | Easy | 0.89 | 0.29 | 1.91 | 0.94 | 0.17 | 0.65 | 0.52 | 0.49 |

this to compute F1 scores between speaker utterances and ground-truth captions over each part of speech. This allows us to evaluate how effectively our speakers identify referents within the target image that a human captioner would consider salient. Higher F1 scores over a part of speech would suggest that a speaker is more capable of identifying concepts that fall under that part of speech.

Images and captions are drawn from the MS COCO dataset introduced in Lin et al. (2014). The train-val-test split given by the MS COCO 2017 dataset is extended to our experimental setup.

## 6.2 EFFECTS OF TOM

We train models equipped with ToM listeners with normal weight (i.e. equally weighting the speaker and the listener scores when reranking output utterances) and with high weight (where the listener weight is set arbitrarily to 1000 and the speaker essentially optimizes for $P(x|u^i)$). In order to isolate the effects of using a ToM listener from the effects of modifying the utterance selection process, we also train a model equipped with a ToM listener that it does not give any weight to. Additionally, we train a model without any ToM component for comparison. We do this for both easy and hard distractors, where the hard distractors were those that were chosen by the hybrid similarity between visual and semantic (CLIP) features outlined in 5. Finally, we give the listener the ground-truth captions over the test set to compute gold-standard metrics of accuracy and fluency.

We find significant performance improvements in Table 1 when speaker models are trained to rerank utterances solely by ToM listener score. Such "high-weight ToM" speaker models achieve accuracy gains of 3.0% and 4.6% on easy and hard distractors, respectively. This suggests that the inclusion of a sufficiently influential ToM reranker during the speaker training process improves speaker performance, although the relative gains appear to be much higher when training on easy distractors.

Table 2: Performance and language features of speakers trained on various distractors. We only show the most performant variants of Caption and Hybrid similarity, with the others shown in A.1.

| Model | Performance | | POS F1 | | | | Average |
| Distractors | Acc | Fluency | ADJ | ADP | NOUN | VERB | Length |
|---|---|---|---|---|---|---|---|
| Base | 0.81 | 1.50 | 0.16 | 0.52 | 0.41 | 0.38 | 8.97 |
| Gold Standard | 0.92 | 2.52 | 1.00 | 1.00 | 1.00 | 1.00 | 10.79 |
| Image | 0.80 | 2.09 | 0.19 | 0.61 | 0.45 | 0.49 | 10.33 |
| Caption | 0.86 | 2.01 | 0.19 | 0.62 | 0.49 | 0.48 | 10.04 |
| Hybrid | 0.85 | 2.19 | 0.20 | 0.62 | 0.49 | 0.47 | 10.18 |

However, we find that speaker models that rerank utterances using a combined speaker-ToM score generally fail to outperform models that do not use their ToM listener in training.

We also find that the usage of a highly-weighted ToM listener leads to significant fluency gains when training on both easy (15.6% relative increase in fluency score) and hard (11.6%) distractors. We also see longer and more complex utterances when using normally or highly weighted ToM listeners. Additionally, we find limited gains in general captioning ability between baseline and high-weight models, as measured by BLEU score. However, these effects are more subtle, and do not always lead to significant accuracy gains, suggesting that the main driver of ToM accuracy gains is increased pragmatic ability. We conclude that usage of a highly influential ToM listener during the training process leads to significant performance and fluency gains. We are also able to qualitatively observe the improvement in model performance from ToM. As seen in one representative example in Fig. 3, our ToM Speaker is able to identify two elements that clearly distinguish the target image from the distractors (i.e. that there are multiple men who are playing baseball) in a fluent utterance.

Finally, we find that the ToM listener successfully approximates the external listener. Models with learned listeners and RSA models with the pretrained listener perform comparably in accuracy and fluency. Because the RSA models represent the upper bound of how good a speaker's listener model can be, this suggests that our learned listeners are very beneficial to the speakers. This is also shown through the high ToM accuracies reported, especially in the most performant models, those with high listener weight. These qualitative and quantitative results provide computational evidence that ToM can play an important role in simulated language acquisition, similarly to how it has been hypothesized to play a critical role in human language acquisition.

## 6.3 Effects of Distractor Difficulty

As shown in Table 2, we generally find significant improvements in language quality in models trained on more difficult distractors. The largest gains are those seen in the fluency score, where difficult distractors achieve gains ranging between 25% to 46%. We also find that models trained on more difficult distractors use more similar vocabulary to the ground-truth captions, as measured by F1 score in the ground-truth captions and utterances produced. This is significantly higher over adpositions, nouns, and verbs on models trained with more difficult distractors. Finally, all speakers trained on difficult distractors generate more complex utterances compared to the base speaker, with utterances that are at least one word longer on average. This supports our hypothesis that when confronted with increased environmental pressure, the speaker adapts by becoming more precise, fluent, and complex with its language. These can be seen qualitatively in one representative example in Fig. 3, which shows that a speaker trained on hard distractors is able to generate more fluent utterances that more precisely describe the image (in this example, correctly identifying an object in the image as a baseball bat, as opposed to an umbrella).

We find smaller differences between the language of models trained with various types of hard distractors. Speaker models trained with visually similar distractors achieve the highest fluency, at 2.094, and form the longest utterances. They also have more precise verb selection, as measured by F1. Speakers trained on distractors that were selected with hybrid or caption similarity, achieved high noun F1 scores of $0.49$ compared to $0.41$ for the base speaker model, indicating that semantically similar distractors in training may be better for identifying salient nouns. We find that training on more difficult distractors does not consistently improve model performance when evaluating on easier distractors. This suggests some disconnect between a language's fluency and its suitability to

the image referential game environment. However, speakers that train on more semantically similar distractors still achieve up to 5 percent higher accuracy than the base speaker, indicating some benefits to performance from training on certain harder distractors.

## 7 RELATED WORK

**Parallels in Human Language Acquisition.** The concept of learning language through repeated exposure to referents is a popular model within the psychology community. Smith & Yu (2008) found that infants resolve the uncertainty of determining which referent in a scene a word refers to by statistical learning over word-scene pairings. Yu & Ballard (2007) incorporated a social element into models by considering "social-cognitive" capacities, such as attention reading, that act as a form of ToM. Yu & Smith (2012) studied caregivers' social impact on early language acquisition using head-mounted cameras, finding that caregiver feedback during moments where a referent was visually dominant directly led to language learning in the infant. Bergelson & Aslin (2017b) found that unrelated images were easier for infants to differentiate between than semantically similar images.

**Pragmatic Modelling.** Andreas & Klein (2016) used pragmatic models to jointly train neural speaker and listener models to play a referential game that involved describing contrasting scenes. They also use a sample-and-rerank method for selecting utterances in this setup. However, they use the actual listener, rather than a learned listener, to rerank utterances. Additionally, we apply this process to a computational model of language acquisition. Nematzadeh et al. (2018) created a dataset to evaluate question-answering models' ability to keep track of inconsistent worldviews. They found that state-of-the-art neural models lack this ability, indicating that they cannot solve tasks with ToM. Monroe et al. (2017) studied the effects of visually similar distractors on a pragmatic model of color identification, finding that pragmatic models had the largest gains in the most difficult settings. Vedantam et al. (2017) also consider the effects of including pragmatic components in image captioning models, namely an internal module to better discriminate between images.

**Emergent Language.** Lazaridou & Baroni (2020) surveyed recent progress in emergent language from multi-agent communication, claiming that further progress can help deep networks become more interactive and interpretable. Lazaridou et al. (2020) place a generally trained language model in a multi-agent environment with task-specific rewards. Similarly to our work, this results in a task-conditional language model which the authors claim can better communicate with humans over visual tasks. Chaabouni et al. (2022) analyzed the effects of task difficulty on emergent communication tasks by varying the number of distractors, leading to negative effects on model performance during evaluation. Mu & Goodman (2021) attempted to improve interpretability of learned languages in referential games by forcing speakers to communicate over sets of objects representing abstract visual concepts, and analyzed the compositionality of the ensuing emergent languages.

## 8 CONCLUSION AND FUTURE WORK

In this paper, we extend an existing computational framework that models the language acquisition process in an image referential game environment. Most notably, we add a ToM component to our speaker models, allowing our speakers to pragmatically rerank candidate utterances with a learned internal listener. We also experiment with increasing distractor difficulty by upweighting more semantically and visually similar distractor images. We find that incorporating ToM into our speaker models leads to improvements in speaker performance. We also find that incorporating harder distractors leads to the development of more complex and fluent languages.

Future work could measure the similarity of the learning process between human learners and our models, and whether the changes implemented in this paper lead to a more humanlike learning process. Further work could also consider the implications of more dynamic difficulty adjustment or curriculum design – for instance, studying whether models trained on a variety of distractor difficulties are able to adjust their utterances to fit a context. Finally, we can study these effects in more complex environments by varying the listener architecture or by considering more difficult settings, such as object referential games. We encourage the machine learning and psychological modelling communities to consider the further incorporation of ToM into computational models of language acquisition, which could help develop more pragmatically-aware models.

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

# A   APPENDIX

## A.1   FULL EFFECTS OF DISTRACTOR DIFFICULTY

In Table 2, we select the caption and hybrid distractors involved in the training of the most performant models to be reported. Here, as promised, we report the results of experiments over a fuller range of distractor variants.

Table 3: Performance and language features of speakers trained on all distractor variants.

| Model Distractors | Performance | | POS F1 | | | | Average |
|---|---|---|---|---|---|---|---|
| | Acc | Fluency | ADJ | ADP | NOUN | VERB | Length |
| Base | 0.81 | 1.50 | 0.16 | 0.52 | 0.41 | 0.38 | 8.97 |
| Gold Standard | 0.92 | 2.52 | 1.00 | 1.00 | 1.00 | 1.00 | 10.79 |
| Image | 0.80 | 2.09 | 0.19 | 0.61 | 0.45 | 0.49 | 10.33 |
| Caption CLIP | 0.83 | 1.87 | 0.18 | 0.57 | 0.46 | 0.43 | 9.56 |
| Caption RoBERTa | 0.83 | 1.93 | 0.20 | 0.60 | 0.48 | 0.47 | 10.06 |
| Caption RoBERTa TFIDF | 0.86 | 2.01 | 0.19 | 0.62 | 0.49 | 0.48 | 10.04 |
| Caption TFIDF | 0.83 | 2.05 | 0.21 | 0.63 | 0.46 | 0.45 | 10.30 |
| Hybrid  CLIP | 0.81 | 1.87 | 0.24 | 0.58 | 0.46 | 0.45 | 9.78 |
| Hybrid  RoBERTa | 0.83 | 1.94 | 0.20 | 0.61 | 0.48 | 0.47 | 10.09 |
| Hybrid  RoBERTa TFIDF | 0.84 | 2.09 | 0.20 | 0.65 | 0.48 | 0.46 | 10.29 |
| Hybrid  TFIDF | 0.85 | 2.19 | 0.20 | 0.62 | 0.49 | 0.47 | 10.18 |

