# OpenReview forum: "Computational Language Acquisition with Theory of Mind"
_ICLR.cc/2023/Conference — ICLR 2023 poster_

### Official Review · Reviewer_J8M3 · 2022-10-23

**Confidence:** 4
**Correctness:** 3
**Technical Novelty And Significance:** 4
**Empirical Novelty And Significance:** 4
**Recommendation:** 8

**Clarity, Quality, Novelty And Reproducibility:**


Quality: The evidence provides support for the conclusions about the impact of adding a ToM component. However there are some shortfalls in the approach that would make the support more clear if addressed.
Clarity: The paper is easy to read, but some pieces of information are missing or vague. The main ideas can be understood.
Novelty: The idea of adding a ToM component to improve langauge adaptation in referential games is novel.
Significance: This may inspire others to improve their language models by adding ToM components. Some may be watching for this sort of experiment (Lazaridou et. al. 2020).
Reproducibility: Some missing details may prevent reproducibility.


**Strength And Weaknesses:**


Strengths
===

* I really appreciate the motivation for this work. Thinking about language acquisition from an evolutionary perspective seems important and the role of Theory of Mind in evolutionary language acquisition in AI has not been explored. This has also been specifically identified as in need of further study in the cited work (Lazaridou et. al. 2020).

* Futhermore, this particular computational relization of Theory of Mind seems new and the evidence points in the direction

* The idea of the listener giving feedback only in certain circumstances, and using the ground truth caption to do that is interesting and potentially novel.


Weaknesses
===

These are roughly ordered by importance.

* (quality) One thing that's still not clear to me is whether the improvement is really because the speaker is modeling the behavior of the listener or whether the ToM listener component is just making it better at image captioning in general. While the improvement of the "Zero" ToM Weight models over the "Baseline" can only be due to a general improvement in image captioning ability, the additional improvement of the "High" ToM Weight models could still be purely due to an additional increase in general image captioning ability. This could be measured by looking at the general captioning quality of the various models. If caption quality stayed the same but listener choice accuracy increased then that would indicate the improvement is really due to the speaker's model of the listener. Even if caption quality went up some component could still be due to the speaker's model of the listener, so I don't think the issue is straightforward to resolve. I think the available evidence points toward some contribution from the ToM component, but more work is needed to resolve this with high confidence.
    * (suggestion) It would certainly help to have a better measurement of the speaker's overall captioning quality. The Part Of Speech F1 and Average Length metrics give some sense of this, suggesting the "High" models are better captioners, but it should be measured with a more established metric like BLEU, SPICE, or CLIPScore[2].

* (clarity, quality, novelty) The connection between ToM and distractor difficulty isn't very clear, and distractor difficulty isn't very well motivated for studying for its own sake. This makes it hard to see why the content of section 6.3 was important to investigate.

* (quality) The listener feedback mechanism is interesting, but the paper does not address how it affects performance in the experiments. How often does the ToM listener receive GT feedback? How does that change over the course of training? This matters because we expect NNs not to be well calibrated by default, and over-confidence could result in this mechanism simply being ignored (with P_max always greater than theta_2 in eq. (2)).

* (clarity) The notation is confusing at times. The reranking weight and caption words both use the letter w with different subscripts. The w_{n^u} notation is never explicitly defined. (Is n the summation variable in eq. 10? Isn't P_ToM a conditional distribution?). I can't understand how the ToM listener is trained because I don't understand eq. 10. Is it supposed to assign high probability to all possible choice images given the caption that the speaker assigns them? Or is it suppose to assign the same label as the actual listener given the choice already made by the listener and the caption the speaker assigns to that choice?

* (quality) The paper does not argue (either empically or rhetorically) whether the diverse choices used in re-ranking are diverse enough for re-ranking to be very meaningful. Often LM samples can be rather redundant and may not always include good choices. Is that the case here?

* (novelty) This work does not cite [1] even though it is very similar. Both works take an image captioning speaker agent and add an internal component which makes the speaker produce utterances that are more discriminative between target and distractor images. I think this work is still different because it is motivated by ToM and thus incorporates feedback from the listener to direct the discriminativity, unlike [1].

* (clarity) RQ1 and RQ2 are not very clear as stated. In RQ1 I see "the performance" of "said models" as referring to their ability to get the other agents to perform well. If it is the listener then perhaps this makes sense, but at this point its not clear whether the listener or the speaker is referred to as "said models" This is also the first mention of RQ2, or at least I'm not immediately sure of how it relates to the intro having been read so far or how it relates to RQ1. A clear definition of what the environment is would be helpful here. Later on there is also a disconnect in flow between section 6.2 and section 6.3, and this would help improve that.

* (clarity) In section 2.1 I is initially referred to as a _set_ of images, while I^N seems to be referencing the same set later. For consistency I^N should be used in the first case or some other adjustment should be made.

* (clarity) "However, the language network parameters are additionally trained [...] to optimize the network values": I don't understand how the listener is trained. Is the COCO train dataset is used?

* (clarity) What exactly are the "easy" distractors? I'm guessing they are randomly selected COCO images, but that should be verified and made clear in the text.

* (clarity) The "Return" metric is not defined or referenced in the results section.



[1]: Vedantam, Ramakrishna et al. “Context-Aware Captions from Context-Agnostic Supervision.” 2017 IEEE Conference on Computer Vision and Pattern Recognition (CVPR) (2017): 1070-1079.
[2]: Hessel, Jack et al. “CLIPScore: A Reference-free Evaluation Metric for Image Captioning.” ArXiv abs/2104.08718 (2021): n. pag.


**Summary Of The Paper:**


This paper shows that inserting a Theory of Mind module into an image captioning agent playing a referential game leads that agent to learn more effective and fluent language.

(motivation)
Some work in developmental psychology says that human can quickly learn and adapt language because they model the mental states of their social companions -- because they have a Theory of Mind (ToM). Recent work has studied image captioning agents that learn or adapt language while playing referential games. This work asks whether adding a ToM component into these agents improves the language they learn.

(approach)
The speaker sees a target image and captions that image. The listener sees the speaker's caption and must use it to pick the correct image from a list of distractor images. The listener also returns the ground truth caption for its selected image when it is confident, but not highly confident. These agents are initialized by pre-training on image-caption pairs. The speaker is finetuned to
1) make the listener guess the correct target (PPO objective) and
2) mimic feedback (GT captions) when given by the listener.

The speaker is also augmented with a ToM component. When the listener gives feedback on an image (potentially not the target) the ToM component is trained to

3) assign high probability to that image given the speaker's caption for it.
During inference potential speaker outputs are re-ranked to choose the utterance which the ToM speaker assigns highest probability.

(evaluation)
Evaluation focuses on the quality of the speaker's language, showing that
1) The listener picks the correct image (Accuracy) more often with the ToM speaker than without. ("Baseline" and "Zero" ToM Weight)
2) The speaker's language is more fluent with ToM (GPT-2 assigns it higher probability)
3) The speaker's language is more similar to ground truth captions (adpositions, nouns, verbs usage; sentence length)
4) Using hard distractors (image choices that are similar to the target) results in better language fluency, though not better Accuracy.

The paper concludes that adding the ToM component improves speaker fluency and effectiveness.


**Summary Of The Review:**


The novelty and potential significance of the work make it a relevant contribution to this community. While the quality and clarity of the work have some flaws, in their current state they are good enough to comprise a meaningful contribution.

---

> ### Author Response · Authors · 2022-11-19
> **Response to Comments from Reviewer J8M3 (2/2)**
>
> > (novelty) This work does not cite [1] even though it is very similar. Both works take an image captioning speaker agent and add an internal component which makes the speaker produce utterances that are more discriminative between target and distractor images. I think this work is still different because it is motivated by ToM and thus incorporates feedback from the listener to direct the discriminativity, unlike [1].
>
> We agree that there are similarities between our work and [1], namely in the investigation of internal pragmatic models in an image captioning speaker agent, and have added it to our Related Work section. However, there are also significant differences between our work and [1], most notably, our work focuses on the process of language acquisition. We also examine the influence of a learned internal listener, and the effects of various distractor image selection methods.
>
> > (clarity) RQ1 and RQ2 are not very clear as stated. In RQ1 I see "the performance" of "said models" as referring to their ability to get the other agents to perform well. If it is the listener then perhaps this makes sense, but at this point its not clear whether the listener or the speaker is referred to as "said models" This is also the first mention of RQ2, or at least I'm not immediately sure of how it relates to the intro having been read so far or how it relates to RQ1. A clear definition of what the environment is would be helpful here. Later on there is also a disconnect in flow between section 6.2 and section 6.3, and this would help improve that.
>
> RQ1 is referring to the speaker; we investigate whether including ToM improves their performance on the image referential game task. We have clarified this, as well as the general role of the environment and the connection between the two questions, in the revised introduction.
>
> > (clarity) In section 2.1 I is initially referred to as a set of images, while I^N seems to be referencing the same set later. For consistency I^N should be used in the first case or some other adjustment should be made.
>
> Thank you for bringing this up! I^N is actually referencing an observation space of N images taken from the set I. We have ensured in the revised submission that we do not use this to refer to the original image set.
>
> > I don't understand how the listener is trained. Is the COCO train dataset used?
>
> We pretrained the listener on the COCO train dataset using mini-batch stochastic gradient descent.
>
> > What exactly are the "easy" distractors?
>
> We can confirm that the “easy” distractors are randomly selected COCO images, and have clarified this in the text.
>
> > The "Return" metric is not defined or referenced in the results section.
>
> The return metric is the average reward (defined in Section 2) over the test dataset. To simplify our results, we have removed the reporting of average reward in Tables 6.2 and 6.3.

---

> ### Author Response · Authors · 2022-11-19
> **Response to Comments from Reviewer J8M3 (1/2)**
>
> We thank the reviewer for their thoughtful feedback and appreciate that they found our submission to be strong.
>
> > One thing that's still not clear to me is whether the improvement is really because the speaker is modeling the behavior of the listener or whether the ToM listener component is just making it better at image captioning in general….If caption quality stayed the same but listener choice accuracy increased then that would indicate the improvement is really due to the speaker's model of the listener…It would certainly help to have a better measurement of the speaker's overall captioning quality. The Part Of Speech F1 and Average Length metrics give some sense of this, suggesting the "High" models are better captioners, but it should be measured with a more established metric.
>
> We have computed BLEU scores for each of our ToM experiment runs, which we include in Table 6.2 of the revised manuscript. Generally, we find that “high” ToM models achieve slightly higher BLEU scores, but the differences are not large. This at least suggests that not all of the gains are coming from increased captioning quality, although we agree that this result needs further investigation.
>
> > The connection between ToM and distractor difficulty isn't very clear, and distractor difficulty isn't very well motivated for studying for its own sake. This makes it hard to see why the content of section 6.3 was important to investigate.
>
> We are generally interested in studying the way in which speakers adapt to their environment in the process of language acquisition. One way in which this happens is through the pragmatic ToM modeling, through which speakers can adapt to their listeners. However, we also consider external pressure from the environment to be part of this story, as more difficult distractors can force the speaker to adapt by developing more precise, fluent language. We have modified our introduction to better motivate the study of distractor difficulty.
>
> > (quality) The listener feedback mechanism is interesting, but the paper does not address how it affects performance in the experiments. How often does the ToM listener receive GT feedback? How does that change over the course of training? This matters because we expect NNs not to be well calibrated by default, and over-confidence could result in this mechanism simply being ignored (with P_max always greater than theta_2 in eq. (2)).
>
> Unfortunately, we did not have time to explore this direction during the rebuttal period. However, Zhu et al (2022), whose language acquisition model we extend, discusses the impact of listener feedback more thoroughly and tunes the $\theta_1$ and $\theta_2$ parameters.
>
> > (clarity) The notation is confusing at times. The reranking weight and caption words both use the letter w with different subscripts. The w_{n^u} notation is never explicitly defined. (Is n the summation variable in eq. 10? Isn't P_ToM a conditional distribution?). I can't understand how the ToM listener is trained because I don't understand eq. 10. Is it supposed to assign high probability to all possible choice images given the caption that the speaker assigns them? Or is it suppose to assign the same label as the actual listener given the choice already made by the listener and the caption the speaker assigns to that choice?
>
> We apologize for the confusing notation and have revised this equation, as well as others identified as confusing in the reviews, to be more clear. We intended to train the internal ToM listener to emulate the actual listener’s scores over time by using the cross-entropy in our revised equation 10 as a loss function. Unfortunately, we inadvertently trained it to give high probabilities to the correct image (see response to R1).  We are attempting to provide a more complete set of results with the revised loss function ASAP.
>
> > (quality) The paper does not argue (either empically or rhetorically) whether the diverse choices used in re-ranking are diverse enough for re-ranking to be very meaningful. Often LM samples can be rather redundant and may not always include good choices. Is that the case here?
>
> While tuning the sampling parameters, we found that higher numbers of samples led to improved performance. This indicates that there was at least some benefit from increased diversity. We did not have time to investigate this further but agree that it is an interesting direction for future study.

---

> ### Author Response · Authors · 2022-12-01
> **Updated ToM Results**
>
> Thank you again for your helpful feedback! As promised, we have rerun our ToM experiments with the revised loss calculation, and have posted the updated results table [here.](https://openreview.net/forum?id=C2ulri4duIs&noteId=xHmIE8MDN1A) We generally found even higher performance in ToM models trained using this loss, and plan to revise our results and discussion accordingly in the camera-ready version of the paper if accepted.

---

### Official Review · Reviewer_NqTJ · 2022-10-24

**Confidence:** 4
**Correctness:** 3
**Technical Novelty And Significance:** 3
**Empirical Novelty And Significance:** 3
**Recommendation:** 6

**Clarity, Quality, Novelty And Reproducibility:**

Clarity
* Mostly very clear.  One question about the material in Table 2

Quality
* High quality focused contribution.

Novelty
* Sufficiently Novel. Tackles a compelling research question from a novel direction.

Reproducibility
* Reasonably described to promote reproducibility. Authors state code and data will be releated.

**Strength And Weaknesses:**

Strengths
* Clear connection from the problem statement to the empirical experimentation
* Clearly written. Relatively easy to understand results and modeling formulation

Weaknesses
* While the findings related to the inclusion of the ToM component are obvious from Table 1, the effects of distractor difficulty in Table 2 are less obvious. It's not obvious how the Model Distractors are connected to the claims of distractor difficulty.  However, the distinction to the Gold Standard are very clear.

**Summary Of The Paper:**

This paper describes an image captioning task to represent language acquisition through its structure.  The learner or "speaker" is the image captioner.  The feedback is provided via a "listener" that provides feedback in the form of a ground truth caption. The innovation comes from the inclusion, in the speaker model of a representation of the listener, the "theory of mind" (ToM) listener.

The authors find that 1) the model learns better with a ToM listener and 2) the outputs are more complex when the feedback is based on difficult distractors rather than easy ones.

**Summary Of The Review:**

This paper describes an approach to evaluate claims of language acquisition through image captioning.  The experimental setting is well motivated and reasonably evaluated.  With the exception of the claims about linguistic complexity being influenced by difficulty of distractors, the findings are clearly evidenced by the experiments.

---

> ### Author Response · Authors · 2022-11-19
> **Response to Comments from Reviewer NqTJ**
>
> We thank the reviewer for their thoughtful feedback and appreciate that they found our submission to be compelling.
>
> > While the findings related to the inclusion of the ToM component are obvious from Table 1, the effects of distractor difficulty in Table 2 are less obvious. It’s not obvious how the Model Distractors are connected to the claims of distractor difficulty. However, the distinction to the Gold Standard are very clear.
>
> While there are still differences between the languages of models trained on different varieties of difficult distractors, the most relevant comparisons in Table 2 are those between models trained on difficult distractors and models trained on the randomly selected base distractors. We have simplified Table 6.3 to better highlight this in our revised submission.
>
> We observed that models trained on difficult distractors are generally (1) more fluent and (2) more precise (as measured by F1, which measures speakers' ability to correctly identify concepts in the image). We see this as a response to the increased similarity between images, which forces models to adapt by learning higher-quality utterances to discriminate between them. This supports our central investigation of how language acquisition models adapt to their surrounding environments.

---

### Official Review · Reviewer_Z8Jq · 2022-10-26

**Confidence:** 3
**Correctness:** 3
**Technical Novelty And Significance:** 2
**Empirical Novelty And Significance:** 2
**Recommendation:** 5

**Clarity, Quality, Novelty And Reproducibility:**

The paper is clearly written until section 3.2. Eqn. 5. O_CG is not defined.

It seems to a straightforward application of adding ToM to the speaker and listener formulations defined in Zhu et al. (2022).

**Strength And Weaknesses:**

Strength
1. Inspired by theory of mind in cognitive science, a critical mechanism of language learning is the ability to infer the mental states of other agents in social environments, the paper presents language-learning agents equipped with ToM.

2. The paper finds that incorporating ToM into the speaker models leads to improvements in speaker performance and incorporating harder distractors leads to the development of more complex and fluent languages.

Weaknesses
1. The problem setting is very limited with vocabulary of 200 words and the max utterance length is 20 tokens. Yet the paper makes very strong claims on answering two fundamental questions:
RQ1. Does the inclusion of ToM in language acquisition models improve the performance and learned language of said models? (internal ToM mechanism)
RQ2. How do our models’ learned languages adapt to discriminate between more visually and semantically similar images? (external environmental pressure)

2. The model design is presented as it is. There is no discussion on alternative designs and whether the design choice achieves the desired goals. For example, why LSTM is used. Would transformer or pretrained language model be a better choice.

**Summary Of The Paper:**

This paper is inspired by cognitive science that young children actively acquire language through interactions with their surrounding environment and caretakers. Specifically one critical mechanism of language learning is the ability to infer the mental states of other agents in social environments, referred to as Theory of Mind (ToM) by Premack & Woodruff (1978). Departing from current approaches of language models,  the paper presents language-learning agents equipped with ToM.  It models ToM by giving the speaker agent an internal listener model that is trained alongside the speaker and using this ToM model to rerank potential utterances. The paper also experimented with varying task difficulty assuming that stronger environmental pressures will promote the development of more complex language. The paper shows that speakers trained with a ToM listener component have higher accuracies than those trained without in our image referential game setting. It further observes that increasing task difficulty in the training process results in more fluent, higher-quality utterances in evaluation.

**Summary Of The Review:**

As it current stands, the paper added theory of mind to the standard speaker and listener formulations. The model design is also very straight forward. The evaluation is on a very limited environment. Why the very limiting image referential game environment is the most relevant for computational language acquisition with theory of mind? What are the difficulties to add ToM to big multi-modal language models such as Flamingo?

In its current form, I do not find the results appealing to the ICLR community.

== post authors' response, and reviewer and area chair discussion
The authors' response is very defensive in nature and not convincing. For example "Additionally, we believe that the image referential environment strikes a balance between ease of investigation and realisticness." Why is the image referential environment strikes a balance and why is it realistic? What other environments are you comparing with?

However, after discussing with other reviewers and the area chair in a virtual zoom meeting, I am a bit more positive on the limited contribution of this paper. I am willing to upgrade my rating to 5: marginally below the acceptance threshold.

---

> ### Author Response · Authors · 2022-11-19
> **Response to Comments from Reviewer Z8Jq**
>
> We thank the reviewer for their thoughtful feedback.
>
> > The problem setting is very limited with vocabulary of 200 words and the max utterance length is 20 tokens. Yet the paper makes very strong claims on answering two fundamental questions…There is no discussion on alternative designs and whether the design choice achieves the desired goals. For example, why LSTM is used. Would transformer or pretrained language model be a better choice.
>
> Following Zhu et al (2022), we limit the vocabulary to 200 words, and the maximum utterance length to 20. These are done due to stability issues encountered when training reinforcement learning systems on large vocabulary spaces. In practice, a large majority of utterances are shorter than our maximum utterance length, meaning the latter is not an especially constraining parameter.
> The main point of the ToM work is that pragmatic modeling with a learned ToM listener can boost speaker performance and utterance quality, which we believe the results using LSTM listeners achieve. However, this is not to say that the model choices made were entirely arbitrary. We did experiment with hyperparameter tuning, as well as alternate utterance selection techniques such as beam search, and found that our models were relatively robust to such changes. Ultimately, we don’t intend to capture all aspects of human language learning, but hope that future work can expand in the direction of creating more realistic language acquisition environments. We have edited the conclusion to better describe how future work can extend our research in such directions.
>
> > The paper is clearly written until section 3.2. Eqn. 5. O_CG is not defined.
>
> Thank you for bringing this to our attention. $O_{CG}$ corresponds to the environment rewards an agent gains from achieving their communicative goals of getting the speaker to identify the correct referent. This has been clarified in the revised draft of our paper.
>
> > The evaluation is on a very limited environment. Why the very limiting image referential game environment is the most relevant for computational language acquisition with theory of mind? What are the difficulties to add ToM to big multi-modal language models such as Flamingo? In its current form, I do not find the results appealing to the ICLR community.
>
> We do not intend to capture all aspects of language learning or prove that our results hold over all types of models, although such results could certainly strengthen this direction in future work. We use PPO-based speakers as we believe that the more “agentic” formulation better represents the process of language acquisition. Additionally, we believe that the image referential environment strikes a balance between ease of investigation and realisticness. Because of the success in this simpler setting, we believe it would be relevant to investigate the inclusion of ToM in more complex models. However, because large language models such as Flamingo can already fluently use language, we consider such models a worse jumping-off point for the investigation of ToM in language acquisition.

---

### Official Review · Reviewer_eJ14 · 2022-10-26

**Confidence:** 4
**Clarity, Quality, Novelty And Reproducibility:** Seems clear enough - I hope authors w…
**Correctness:** 4
**Technical Novelty And Significance:** 3
**Empirical Novelty And Significance:** 3
**Recommendation:** 6

**Strength And Weaknesses:**

# Strengths

This paper presents a number of interesting results in a simulated language learning environment and explores a number of sensible hypotheses about how modeling and dataset choices change the learned languages (though several of these have been explored already - see Weaknesses below). I think the most interesting contribution of this paper (which is highlighted by the title) is the learned ToM-style internal listener model.

Authors are correct that in a lot of the rational speech acts literature, the speaker is allowed to explicitly optimize against the known parameters/probabilities of the partner listener model. Showing that speakers can build internal model of listeners from a weaker supervision signal (just listener choices in certain contexts), and still use this to improve generation, is an interesting finding and likely to be useful to the broader community.

# Weaknesses

The weaknesses I see in the paper have to do with missing contextualization with related work (most salient), a confusion about some of the modeling details (equation 10), as well as some lesser concerns about the broader applicability of this work as a whole.

## Missing contextualization with related work

I believe this paper has more similarities to related work than it lets on. One clear lacuna is discussion with Rational Speech Act models of pragmatic language use. In particular, the idea of an internal listener model that is used to rerank utterances is precisely the "pragmatic speaker" formulation in RSA which optimizes utility of an internal literal listener. This idea was precisely implemented in [Andreas and Klein, 2016](https://arxiv.org/abs/1604.00562) which authors correctly note. But the similarities are much more than that: Andreas and Klein precisely propose "sample and rerank RSA" where candidate utterances are sampled from a speaker model, then rereanked with an internal listener module. This process is *identical* to the ToM speaker, the only difference being that the ToM listener is learned. Andres and Klein also define the same tradeoff between optimizing for the listener probability vs optimizing the speaker probability of utterance. I think these similarities should be highlighted more beyond just "we draw inspiration from." Another relevant paper is [Amortized RSA](https://arxiv.org/abs/2006.00418) which trains a model directly to maximize likelihood via gumbel-softmax (similar to the strictly communicative objective in section 3.1 of this paper, though here PPO is used as the optimizer). Again this is with a fixed listener model; I think the current paper should more clearly highlight the idea of using a learned listener model as the key differentiator over existing work in data-driven RSA.

The idea of increasing difficulty of reference games by making distractors more similar is also a common idea in the literature, e.g. this experimental condition has been explored in [Monroe et al. 2017](https://aclanthology.org/Q17-1023/), [Achlioptas et al., 2019](https://arxiv.org/abs/1905.02925) etc, and indeed show positive effects on the pragmatics and complexity of language when trained in "harder" games.

Another relevant paper which should be discussed is [Lazaridou et al., 2020](https://arxiv.org/abs/2005.07064) which examines the idea of initializing a pretrained image captioning model, then finetuning it with a communicative reward, similar to the present study, as well as using the listener model (jointly learned with the speaker) to improve generations. Note that they (and many in the literature) observe this phenomenon of semantic drift, i.e. speakers begin to use language in ways not implied by the original static training dataset. Did authors find this phenomenon here?

## What's happening in Equation 10?

Equation 10 (ToM objective) and its description are confusing to me. Equation 9 shows $P_{ToM}(I_t \mid u^j)$ playing the role of the listener, i.e. a probability distribution over images $I_t$ given utterances $u^j$. But here in Eq 10 we are evaluating $P_{ToM}(w_u^n)$. What does this mean? What is $w_u^n$? (Seems overloaded with the ToM hyperparameter $w_l$). What is the summation index variable? Description says the ToM objective is "defined as the cross-entropy loss between the distribution of the ToM listener and that of the speaker". Should "speaker" be "true listener" here? This is more aligned with "untrained listener trained to emulate the actual listener's outputs over time."

I have a mental model of what is going on here (that we are training the ToM listener to mimic the actual listener, whenever we actually get listener choices), but I'm not sure if this is actually happening as I found this section difficult to parse. It's possible I'm misunderstanding something here, and I can't be too confident in my recomendation until this is cleared up.

## Unclear whether this is a realistic setting for either NLP/CogSci communities

This paper certainly has some interesting experiments and modeling contributions, but one weakness I see (and I think this is true of many computational simulations of language acquisition, e.g. emergent communication) is the degree to which this is a particularly useful simulation for either (1) giving us better NLP systems, or (2) teaching us about language development. Re: (1), it would be great if authors gave us more concrete settings or potential applications where training a speaker model in this kind of fashion, with natural language feedback, might be useful for building better NLP systems. Re: (2), there are a lot of ad-hoc hyperparameter choices in this simulation environment (e.g. the thresholds at which listeners choose to abstain from providing feedback, or provide the ground-truth caption; the various hyperparameter tradeoffs between speaker objectives) that could drastically change results. If results are fairly sensitive to such ad hoc hyperparameter choices, then we should be careful to make broad sweeping conclusions about what this tells us about human and/or machine language development.

Moreover, it's a little unclear how authors intend, in the conclusion, for this study to support further investigations into the "similarity of the learning process between human learners and our computational models." Babies learn in naturalistic environments, certainly aren't initialized with ResNets, and do far more than just play referring expression games.

I don't view this as a fatal weakness of the paper - I think these are fundamental problems that a lot of papers in this kind of line have, and this debate need not be resolved here.

## Other minor considerations

- It would be great to have an explicit measure of the ToM model success, i.e. how well does the ToM listener approximate the real listener throughout training? I'm interested in a more isolated metric for measuring this beyond just "the full ToM speaker does better". If we get very close to the true listener then it makes perfect sense that ToM modeling would improve generation.
- How much is the "learning from feedback" objective required to get learning off the ground in this setting? Assuming the LSTM is not pre-initialized (which I don't think it is), It seems extremely difficult to learn a good NLG policy from scratch given the massive complexity of the natural language search space. What happens when this objective is turned off? How often is natural language feedback given?
- I find the use of "learning from feedback" to be a bit misleading compared to how the term "feedback" has been traditionally used in the literature (e.g. [1](https://arxiv.org/abs/2204.14146), [2](https://arxiv.org/abs/2009.14715)) - "feedback" implies some language corrections or explanation for why a referring expression is correct/incorrect, but in reality this is just providing the ground-truth label to the agent (which you then optimize for explicitly).
- Notation is a little convoluted at times, e.g. eq 7 $w_{u^j_1}$, is it really necessary to have these multiple nested subscripts?
- I think the conclusion statement that "psychological modelling communities...consider the further incorporation of ToM into language learning simulations" may be slightly uncouth, given that theory of mind is a central idea in an extremely long line of literature in both computational and non-computational pragmatics dating all the way back to Grice.
- I would advise authors to be careful about what constitutes "better" language, and claims that the speakers trained with harder distractors have strictly higher obejctive measures of "utterance quality". If we adopt a fully pragmatic view of language, then the best language should be as concise as possible given the context, and with simpler distractors, it makes sense that the language is simpler - that is the world they are trained on, after all. It'd be interesting to see whether speakers trained on a mixture of context difficulties can modulate the complexity of their utterances when the context demands it.

**Summary Of The Paper:**

This paper presents a study of "computational language acquisition", where a speaker agent learns to use natural language to play Lewis-style referential games. These games are played with a fixed, listener "teacher" agent which makes decisions based on the referring expressions generated by the speaker, optionally giving "natural language feedback" in terms of the ground-truth caption.

This setting has been explored in existing work; the authors' main contribution is to outfit the speaker with a "theory of mind" listener model, which is a learned model of the listener given listener feedback over interactions. Then the speaker model looks like a pragmatic speaker similar to the Rational Speech Acts literature, where it optimizes a weighted objective of (1) truthfulness (i.e. probabilty under the image captioning model) and (2) listener utility under the learned ToM model. Authors show that, for some combination of ToM speaker hyperparameters, ToM speakers are able to indeed generate utterances at higher quality than a speaker without ToM.

Some secondary analyses are also conducted on whether the difficulty of distractors sampled from reference games changes the learned languages; authors find they do, in that more difficult games result in better languages according to various metrics (e.g. utterance complexity, length, closeness to ground-truth caption).

**Summary Of The Review:**

To summarize, I think this paper presents a technically sound set of experiments, although many of the findings here are quite similar to findings that already exist in the computational pragmatics literature. Another weakness is the several seemingly ad-hoc hyperparameter design choices that go into the environment on which the authors base their conclusions.

However, the idea of learning an internal listener model for a pragmatic speaker, is, to my knowledge, new, and is likely to be of interest to researchers in this space. Based on this, I am marginally in favor of acceptance, though I don't (at the moment) feel particularly strongly and look forward to the other reviewers and the author response.

---

> ### Author Response · Authors · 2022-11-19
> **Response to Comments from Reviewer eJ14 (4/4)**
>
> > I would advise authors to be careful about what constitutes "better" language, and claims that the speakers trained with harder distractors have strictly higher obejctive measures of "utterance quality". If we adopt a fully pragmatic view of language, then the best language should be as concise as possible given the context, and with simpler distractors, it makes sense that the language is simpler - that is the world they are trained on, after all. It'd be interesting to see whether speakers trained on a mixture of context difficulties can modulate the complexity of their utterances when the context demands it.
>
> The point that simpler distractors lead to a simpler resulting language is correct and well-received. However, we still believe that the converse of this statement - that more difficult distractors force models to adapt by creating more discriminative captions - is an important theme in our work. While we agree that conciseness is important in a pragmatic view of language, we still think that the fluency and precision of utterances trained on more difficult distractors are noteworthy, especially given the low quality of the baseline utterances. We are intrigued by the ability of pragmatic speakers to modulate the complexity of utterances based on the surrounding context, and believe that such analysis could significantly contribute to this research direction. Unfortunately, it may have to be explored in future work due to time constraints.
>
> [1]: Vedantam, Ramakrishna et al. “Context-Aware Captions from Context-Agnostic Supervision.” 2017 IEEE Conference on Computer Vision and Pattern Recognition (CVPR) (2017): 1070-1079.
>
> [2] Elika Bergelson and Richard Aslin. Semantic specificity in one-year-olds’ word comprehension. Language Learning and Development, 13:1–21, 06 2017a. doi: 10.1080/15475441.2017. 1324308.
>
> [3] Landau, B., Smith, L. B., & Jones, S. S. (1988). The importance of shape in early lexical learning. Cognitive Development,3(3), 299–321.

---

> ### Author Response · Authors · 2022-11-19
> **Response to Comments from Reviewer eJ14 (3/4)**
>
> > Moreover, it's a little unclear how authors intend, in the conclusion, for this study to support further investigations into the "similarity of the learning process between human learners and our computational models." Babies learn in naturalistic environments, certainly aren't initialized with ResNets, and do far more than just play referring expression games.
>
> While our image referential environment is simpler than real-world language acquisition settings, we would like to note that comparable experiments in the field of experimental psychology also intentionally simplify their settings in an attempt to get crisp conclusions. For instance, Bergelson and Aslin (2017b) use an image referential game to investigate human infants’ knowledge of familiar nouns. This is similar to the computational setting in which we train our agents.
>
> Additionally, the speaker’s usage of a pretrained ResNet to create image embeddings may be somewhat simplified, but at the same time, it also mirrors how babies develop perception and world modeling before language activity. For example, [3] shows that infant word learning is heavily influenced by shape bias, suggesting that perception is sufficiently developed in infants to play an important role in language acquisition. We agree that it could be interesting for future work to separately train the speaker’s image embeddings, but we would argue that it is probably more realistic to model a language learner as having some baseline perception ability.
>
> > It would be great to have an explicit measure of the ToM model success, i.e. how well does the ToM listener approximate the real listener throughout training? I'm interested in a more isolated metric for measuring this beyond just "the full ToM speaker does better". If we get very close to the true listener then it makes perfect sense that ToM modeling would improve generation.
>
> Following this suggestion, we have computed ToM listener accuracy (how often the ToM listener correctly identifies what choice the actual listener will make). We find that the ToM listener generally does a reasonable job of making these predictions, with accuracy in the mid-80s for our learned listener models. In the revised manuscript, we include this information in Table 6.2 with the other ToM results.
>
> > How much is the "learning from feedback" objective required to get learning off the ground in this setting? Assuming the LSTM is not pre-initialized (which I don't think it is), It seems extremely difficult to learn a good NLG policy from scratch given the massive complexity of the natural language search space. What happens when this objective is turned off? How often is natural language feedback given?
>
> We do not explore this in our paper; however, Zhu et al (2022) do study the relative importance of the communicative goals and linguistic input mechanisms. They find that while feedback drives the formation of more fluent utterances, communicative goals drive referential game performance and language learning. The environment constraints also help facilitate learning, which the reviewer correctly notes can be difficult in such a complex space.
>
> > I find the use of "learning from feedback" to be a bit misleading compared to how the term "feedback" has been traditionally used in the literature (e.g. 1, 2) - "feedback" implies some language corrections or explanation for why a referring expression is correct/incorrect, but in reality this is just providing the ground-truth label to the agent (which you then optimize for explicitly).
>
> We interchangeably use “linguistic input” and “feedback” to describe what the listener communicates to the speaker, but recognize that this can lead to confusion, and have tried to clarify discussion of these terms in the revised manuscript.
>
> > I think the conclusion statement that "psychological modelling communities...consider the further incorporation of ToM into language learning simulations" may be slightly uncouth, given that theory of mind is a central idea in an extremely long line of literature in both computational and non-computational pragmatics dating all the way back to Grice.
>
> We definitely do not intend to overstate the importance of our work or to overlook the past work in pragmatics that we are most certainly building off! However, as we stated above, we do think that our specific application of Theory of Mind to simulations of **language learning** is novel, and tried to highlight this particular aspect more in our revised discussion.

---

> ### Author Response · Authors · 2022-11-19
> **Response to Comments from Reviewer eJ14 (2/4)**
>
> > Equation 10 (ToM objective) and its description are confusing to me…Description says the ToM objective is "defined as the cross-entropy loss between the distribution of the ToM listener and that of the speaker". Should "speaker" be "true listener" here? This is more aligned with "untrained listener trained to emulate the actual listener's outputs over time."
>
> Thank you for catching this! Your comment led us to reinvestigate the exact implementation of the ToM listener in the code. Your intuition for what should be happening is exactly correct (i.e. emulating the true listener’s actions), it appears that while creating multiple versions of our models for ablations, we inadvertently commented out this training component. As a result, we actually presented results for the case where the ToM listener is trained to select the correct image given the speaker’s utterance.
>
> Despite this, our ToM listeners still predict the actual listener’s choices with reasonable accuracy. However, this is still a strictly weaker result than what we intended to show, and we have therefore begun training the corrected version for a full comparison. Unfortunately, that result will take longer to complete so we cannot include it just yet, but hope to have that in a follow-up response and will include the full set of ablation comparisons in the camera-ready submission.
>
> > It would be great if authors gave us more concrete settings or potential applications where training a speaker model in this kind of fashion, with natural language feedback, might be useful for building better NLP systems.
>
> We thank the reviewer for this point. We would like to emphasize that immediately training better NLP models is not the main point of our work, as we are primarily interested in the question of what effects theory of mind has on language acquisiton in computational models. This is similar to a large number of preceding works in the field of computational pragmatics ([1] is just one example). However,  there is interesting future work that could be done in this area.
>
> There are a few cases where this kind of training could improve NLP systems. For instance, the conjecture that this joint training could be used to better train pragmatics models that imbue NLP systems with contextual knowledge. One potential application of this is on joint language tasks with human partners. In this case, this type of speaker training could allow a model to develop a personalized pragmatic model of its human partner without significant pretraining.
>
> > Re: (2), there are a lot of ad-hoc hyperparameter choices in this simulation environment (e.g. the thresholds at which listeners choose to abstain from providing feedback, or provide the ground-truth caption; the various hyperparameter tradeoffs between speaker objectives) that could drastically change results. If results are fairly sensitive to such ad hoc hyperparameter choices, then we should be careful to make broad sweeping conclusions about what this tells us about human and/or machine language development.
>
> In terms of the hyperparameters mentioned above, the listener thresholds are taken from Zhu et al (2022), who use grid search to tune them. We also informally tuned the objective weights, as well as other hyperparameters such as the number of samples considered, and found that model performance was generally robust to changes in these hyperparameters. We plan to conduct a more rigorous hyperparameter search in the future, but unfortunately did not have time to do so during the rebuttal period.

---

> ### Author Response · Authors · 2022-11-19
> **Response to Comments from Reviewer eJ14 (1/4)**
>
> We thank the reviewer for their thoughtful feedback and appreciate that they found our submission to be compelling. We definitely want to avoid overextending our claims to general language usage models. Unlike previous work, we view not only the learned model of RSA, but also the application of ToM and distractor difficulty to **language acquisition**, to be an important differentiating factor between this work and some of the related work mentioned.
>
> > I believe this paper has more similarities to related work than it lets on. One clear lacuna is discussion with Rational Speech Act models of pragmatic language use…Andreas and Klein precisely propose "sample and rerank RSA" where candidate utterances are sampled from a speaker model, then rereanked with an internal listener module. This process is identical to the ToM speaker, the only difference being that the ToM listener is learned. Andreas and Klein also define the same tradeoff between optimizing for the listener probability vs optimizing the speaker probability of utterance. I think these similarities should be highlighted more beyond just "we draw inspiration from."
>
> Thank you for raising this concern, and we agree that this similarity should have been better highlighted in our paper. In the revised version of our manuscript, we revised our Related Work section to more clearly and accurately discuss these similarities. However, we do view the main arguments of this paper as related to effects on language learning, rather than the pragmatic model architecture itself. While the main difference between our internal listener and that of work such as Andreas and Klein is that our listener is learned, our application of theory-of-mind informed models to an RL-based language acquisition setting is also novel.
>
> > Another relevant paper is Amortized RSA which trains a model directly to maximize likelihood via gumbel-softmax (similar to the strictly communicative objective in section 3.1 of this paper, though here PPO is used as the optimizer). Again this is with a fixed listener model; I think the current paper should more clearly highlight the idea of using a learned listener model as the key differentiator over existing work in data-driven RSA.
>
> In order to more clearly emphasize the difference between this learned listener model and existing data-driven RSA, we also added additional experiments to section 6.2. In particular, we compared the performance of speakers that use learned listeners (our original model) and the ground truth listener (representing RSA models such as that of Andreas). We find that our model accuracy with a learned listener is only slightly lower than the upper-bound performance of RSA models with the actual listener. The difference is small on both easy and hard distractors, suggesting that our learned listeners do a reasonable job of approximating the actual listener.
>
> > The idea of increasing difficulty of reference games by making distractors more similar is also a common idea in the literature, e.g. this experimental condition has been explored in Monroe et al. 2017, Achlioptas et al., 2019 etc, and indeed show positive effects on the pragmatics and complexity of language when trained in "harder" games.
>
> Thank you for raising this point as well. We have revised our Related Work section to more thoroughly cover these works. However, we have not found significant past work that specifically considers the impact of distractor difficulty on models of language learning, and still believe this aspect of our work to be novel.
>
> > Another relevant paper which should be discussed is Lazaridou et al., 2020 which examines the idea of initializing a pretrained image captioning model, then finetuning it with a communicative reward, similar to the present study, as well as using the listener model (jointly learned with the speaker) to improve generations. Note that they (and many in the literature) observe this phenomenon of semantic drift, i.e. speakers begin to use language in ways not implied by the original static training dataset. Did authors find this phenomenon here?
>
> Zhu et al (2022), whose model we extend, find that speakers tend to “overextend” concepts to describe objects not originally covered by the concept in the training data. However, due to the strong feedback mechanisms in our speaker formulation, it is unclear if this is actual semantic drift or just speaker inaccuracy. Unfortunately, we did not have time to further study this during the rebuttal period, but thank the reviewer for raising the point, and consider this an interesting avenue of exploration for future work.

---

> ### Author Response · Authors · 2022-12-01
> **Updated ToM Results**
>
> Thank you again for your helpful feedback! As promised, we have rerun our ToM experiments with the revised loss calculation, and have posted the updated results table [here.](https://openreview.net/forum?id=C2ulri4duIs&noteId=xHmIE8MDN1A) We generally found even higher performance in ToM models trained using this loss, and plan to revise our results and discussion accordingly in the camera-ready version of the paper if accepted.

---

### Author Response · Authors · 2022-12-01
**Updated ToM Results**

Thank you again to all the reviewers for your helpful feedback; we hope that our responses and changes made to the paper have strengthened our work. As promised in our responses to Reviewers eJ14 and J8M3, we have rerun our experiments with the corrected ToM loss calculation, the results of which we share below:

\\begin{array} {|r|r|}\\hline \\hline \\text{ToM Weight} & \\text{Distractors} & \\text{Accuracy} & \\text{BLEU} & \\text{Fluency} & \\text{ToM Acc} & \\text{ADJ F1} & \\text{ADP F1} & \\text{NOUN F1} & \\text{VERB F1} \\\\ \hline \\text{Baseline (No ToM)} & \\text{Easy} & 0.81 & 0.20 & 1.50 & \\text{N/A} & 0.16 & 0.52 & 0.41 & 0.38 \\\\ \\text{Baseline (No ToM)} & \\text{Hard} & 0.81 & 0.24 & 1.87 & \\text{N/A} & 0.24 & 0.58 & 0.46 & 0.45 \\\\ \\text{Gold Standard} & \\text{N/A} & 0.92 & 1.00 & 2.52 & \\text{N/A} & 1.00 & 1.00 & 1.00 & 1.00 \\\\ \\hline \\text{Zero} & \\text{Hard} & 0.83 & 0.26 & 1.99 & 0.81 & 0.22 & 0.64 & 0.49 & 0.47 \\\\ \\text{Normal} & \\text{Hard} & 0.85 & 0.26 & 2.25 & 0.88 & 0.22 & 0.65 & 0.52 & 0.49 \\\\ \\textbf{High} & \\textbf{Hard} & \\textbf{0.88} & \\textbf{0.27} & \\textbf{2.23} & \\textbf{0.89} & \\textbf{0.22} & \\textbf{0.66} & \\textbf{0.52} & \\textbf{0.50} \\\\ \\text{High RSA} & \\text{Hard} & 0.87 & 0.28 & 2.26 & 0.93 & 0.23 & 0.65 & 0.50 & 0.49 \\\\ \\hline \\text{Zero} & \\text{Easy} & 0.85 & 0.25 & 1.73 & 0.85 & 0.21 & 0.57 & 0.48 & 0.49 \\\\ \\text{Normal} & \\text{Easy} & 0.88 & 0.26 & 2.09 & 0.91 & 0.21 & 0.64 & 0.50 & 0.52 \\\\ \\textbf{High} & \\textbf{Easy} & \\textbf{0.88} & \\textbf{0.27} & \\textbf{2.07} & \\textbf{0.91} & \\textbf{0.22} & \\textbf{0.65} & \\textbf{0.51} & \\textbf{0.50} \\\\ \\text{High RSA} & \\text{Easy} & 0.89 & 0.29 & 1.91 & 0.94 & 0.17 & 0.65 & 0.52 & 0.49 \\\\ \\hline  \\end{array}

We find that these corrected results reinforce our claims, and actually show higher gains from including ToM than previously stated. ToM speakers trained on easy distractors with high listener weights outperform the baseline by seven percent and outperform speakers trained with a listener weight of zero by three percent. We observe similar results on ToM speakers trained on hard distractors with high listener weights, as well as stronger results from ToM speakers trained with normal listener weight.

We also observe similar results in terms of utterance quality. ToM speakers outperform both baseline and zero-weight speakers in F1 scores, BLEU scores, and fluency. Finally, as expected, we find that we learn better approximations of the actual listener after correcting this loss issue. Our corrected ToM listener models correctly predict the listener’s actions approximately ninety percent of the time. We plan to include our revised results and discussion in the camera-ready version of our paper if accepted.

---

### Decision · Program_Chairs · 2023-01-20

**Decision:**

Accept: poster

**Justification For Why Not Higher Score:**

See reviewer discussion above

**Justification For Why Not Lower Score:**

I think overall the paper presents interesting ideas and although there are concerns around the experimental setup being too simple, I think the findings could help future work in this direction. If there is space in the proceedings, I recommend acceptance but the decision can be bumped down.

**Metareview: Summary, Strengths And Weaknesses:**

This paper develop language-learning agents equipped with a theory-of-mind (ToM) module. They consider referential games, where a speaker agent learns to use language to communicate to a listener that has to pick a target image only known to the speaker. As pointed out by reviewer eJ14, this setting has been explored in prior work, but the key contribution of this paper is to add a ToM listener simulator within the speaker, where the simulated listener model is learned using listener feedback in interactions. This allows the speaker to optimize both truthfulness and listener utility and generate appropriate utterances.

Experiments are mixed but some variants of the method work better than baselines that do not learn a listener model using ToM. The paper also performs some analyses that may inform future work in this direction. The main criticisms of this paper were the fact that the vocabulary is quite limited (200 words) and a max utterance length of 20 words (Z8Jq) and missing contextualization with related work (eJ14). The authors promised to resolve the latter with revisions to the paper. If accepted, I strongly encourage the authors to tone down the claims in the abstract and introduction to better reflect scope of the actual empirical results in the paper (and mention upfront the limited scale of the experiments).

**Note From Pc:**

if the above contains the word "oral" or "spotlight" please see: "oral" presentation means -> notable-top-5% and "spotlight" means -> notable-top-25%. As stated in our emails, we are disassociating presentation type from AC recommendations

**Summary Of Ac-Reviewer Meeting:**

Reviewer eJ14 was not present, but sent an email summary of their thoughts after the author response. Overall, their rating of the paper slightly decreased due to the points raised by Z8Jq and other reviewers, but they were on the fence.

Meeting summary:
All the reviewers agreed the paper has an interesting idea, but the key weakness brought up was the issue of the setup being too simple (200 word vocabulary). The discussion centered around whether this method would scale to more realistic settings and as a result, whether the claims of the paper are too strong compared to the actual results. Another point raised was the mistake in equation 10, which seems to have been a bug that the authors caught and fixed. This seems to have improved the results slightly, but there were questions raised around the soundness of the approach as a result. However, some reviewers found the fix to be a positive result of the review process. At the end of the discussion, Z8Jq was still not convinced but suggested they were probably at the level of borderline reject.